# An Outlook on Dental Practices to Avoid the Oral Transmission of COVID-19

**DOI:** 10.3390/microorganisms11010146

**Published:** 2023-01-06

**Authors:** Manal Alsulami, Waad Kattan, Lama Alsamadani, Ghadah Alahmari, Wasan Al Juhani, Maha Almabadi

**Affiliations:** 1Basic Sciences Department, Vision Medical College, Jeddah 23643, Saudi Arabia; 2Dentistry Department, Vision Medical College, Jeddah 23643, Saudi Arabia; 3Board Gaurad Medical Center, Riyadh 14214, Saudi Arabia; 4Health Bracelet Dental Complex, Tabuk 47914, Saudi Arabia

**Keywords:** COVID-19, oral microbiome, oral hygiene, virus transmission, dental practices, dental clinics

## Abstract

The oral microbiome plays an important role in the maintenance of immune homeostasis, whereas its association with SARS-CoV-2 infection remains under investigation. Since the oral path is one of the transmission routes for COVID-19, we attempt to show the relationship between the oral microbiome, COVID-19 infection, and oral hygiene. We highlight the importance of oral hygiene to control the infection, especially for ICU cases with COVID-19. Moreover, we present the current strategies adapted by in-person dental clinics to overcome the spread of COVID-19. New emerging policies and protocols suggested during the pandemic and their future implementation to minimize virus transmission are also summarized in this review.

## 1. Introduction

Corona Virus Disease 2019 (COVID-19), or severe acute respiratory syndrome coronavirus 2 (SARS-CoV-2), is caused by a novel coronavirus that started in Wuhan, China in December 2019 [1,2] and rapidly spread all over the world. Th World Health Organization (WHO) declared COVID-19 a global pandemic in March 2020, and it resulted a prolonged lockdown, economic crisis, and caused approximately 6,656,601 deaths globally by 12 December 2022 [2,3,4]. The common symptoms of the disease are fever, headache, dry cough, sore throat, and sneezing, which appear after COVID-19′s incubation period of 2–14 days (median, 4 days) and have changed with virus variants over time [3]. The intensive research literature of the last two years shows the cause, transmission route, virus variants, cell–virus interaction pathways, and their effects on the human body. Since SARS-CoV-2 is a pathogenic virus, the lungs are the primary organs that become exposed to the virus and are damaged if not treated urgently and properly. Several pathways of virus–human cell interaction have been highlighted in recent research; however, the mechanism of how the virus interacts with the microorganisms present in the lungs is still under investigation. Human lungs and airways harbor diverse microbial compositions which could be changed during severe respiratory infections such as COVID-19. Modern metagenomic and next-generation sequencing (mNGS) techniques helped in identifying lung-microbiome diversity in health and disease. Unlike the gut microbiome, the lungs microbiome is more dynamic and transient because of its bidirectional connection with the oral microbiome through the continuous movement of air and mucus exchange between two organs. A recent metatranscriptome sequencing to determine types of bacteria present in the bronchoalveolar lavage fluid (BALF) of eight COVID-19 patients revealed the presence of high levels of commensal bacteria in the oral and upper respiratory track compared to the control sample [5], indicating a significant relationship between the oral and upper respiratory track microbiomes in health and disease.

At the beginning of the COVID-19 pandemic, it was mentioned that a major transmission route of the COVID-19 virus is the nasal cavity, and oral manifestation is not involved [6]. However, recently, it has been reported that oral microbiomes and lung microorganisms can exchange bacteria and viruses via the aspiration axis. It is considered a key factor leading to many infectious diseases such as flu and strep throat [7]. Some studies mentioned that SARS-CoV-2 was detected in the saliva of COVID-19 patients, providing evidence of the presence of the SARS-CoV-2 virus in the oral cavity, with saliva being used for PCR tests for the screening of COVID-19 patients during the pandemic [3,6].

Some studies show that there are two routes of transmission for SARS-CoV-2: (1) oral and (2) nasal cavities [8,9,10,11]. The oral cavity is the main way for the ingress of SARS-CoV-2 into the human body. It is a portal to the trachea, bronchi, alveoli, and lungs, which are inflamed by viral infection. Scientists found that angiotensin-converting enzyme 2 (ACE2), a host receptor for SARS-CoV-2, is expressed on the tongue and other oral tissues [10]. As a result, the oral cavity is considered as the site of virus replication and proliferation. Among several diverse symptoms, COVID-19-positive patients suffered from loss of taste and experienced tongue pain, which might be associated with the high expression levels of ACE2 in the tongue epithelial cells [10]. Moreover, researchers reported that there is an opportunity for the aspiration of oral bacteria or oral viruses within saliva into the lower respiratory tract, which might be a complicating factor for COVID-19 (Figure 1) [11,12]. Chronic diseases such as obstructive pulmonary disease (OPD) and diabetes are identified as COVID-19 comorbidities. OPD and diabetes have a strong association with chronic periodontitis and periodontal pathogens [13]. Oral hygiene is a key component to avoiding these comorbidities. Additionally, oral bacteria, especially periodontal pathogens, might be a proinflammatory stimulus to respiratory epithelia upon its contact with aspirated bacteria. Therefore, oral hygiene is important, especially during the pandemic (COVID-19), and everybody should be aware of oral health care and diseases because it could prevent SARS-CoV-2 from occurring [13].

Moreover, the nasal cavity is considered an entry point for COVID-19, and it shares the same passage as the oral cavity until reaches the lung (Figure 1). The nasal cavity may play a major role in viral transmission through contaminated airborne droplets [14,15]. COVID-19 is mainly spread via the respiratory route after the inhalation of contaminated droplets or particles. Generally, transmission of SARS-CoV-2 occurred by breathing or direct contact with virus-containing droplets and aerosols from infected people through coughing and sneezing [14,15]. As a result, when the viruses reach the nasal cavity, it moves into the host epithelial cells through the angiotensin-converting enzyme 2 (ACE2) receptor, which is prominently presented in epithelial-cell linings in the respiratory and digestive tract systems. In addition, the virus might exploit existing secretory pathways in nasal epithelial cells, because it does not need cell lysis for release [14,15]. The aim of this review is to show the relationship between the oral microbiome, SARS-CoV-2 infection, and oral hygiene, and the current strategies that have been recommended by the World Health Organization (WHO) and American Dental Association (ADA), to control COVID-19 in the dental clinic.

## 2. A Three-Dimensional Relationship between Oral Microbiomes, COVID-19 Infection, and Oral Hygiene

The oral microbiome resides in the oral cavity and co-evolves with the host tissues. This is controlled by a bidirectional interaction between the microbiome and the host. For instance, some bacterial species within the oral cavity, such as strains of *Streptococcus* (*S.*) mutans, can stop the invasion and the potential growth of harmful endogenous microbial agents by producing antimicrobial peptides known as bacteriocins [9]. Bacteria in the oral cavity are systematized in a structure called “biofilm”, which is a complex protecting structure in which bacterial groups exist in an extracellular matrix to protect themselves from external agents [9]. Therefore, oral bacteria provide the best environment to increase the rate of their growth within the oral cavity, as well as increasing the rate of pathogenic bacteria which lead to oral diseases. As a result, an immune system with intensive oral health care keeps the pathogenic bacteria under control and prevents problems such as tooth decay and gingivitis.

Saliva plays a critical role in preserving oral health and controlling healthy oral microbiota since it has organic and inorganic compounds. It provides an acquired hard glossy substance which covers the crown of a tooth, called enamel, and mucosal, which are the basis for the initial colonization by the microorganisms. In addition, saliva dilutes and removes microorganisms and dietary components from the mouth. It maintains the microbial environment via the antimicrobial action of particular proteins, such as peroxidase, and lysozyme [16]. That is why people (older adults) who have a low level of saliva production seem to have a high risk of getting dental caries [17].

Recent studies show a prevalence of oral bacterium within the oral cavity of healthy individuals such as Firmicutes, Proteobacteria, Bacteroidetes, and Spirochaetes [16,17,18], whereas they found less abundant fungal such as microbes of the genera Candida. The study investigates the variety in the composition of the bacterial oral community among ten healthy individuals [16,17,18]. It confirms that 15 bacterial classes were conserved between all individuals, with significant differences at the species and strain level. Moreover, sometimes, fungi, archaea, and viruses alter the environment of the oral cavity through their interactions with biofilms, which maintains health or changes the stability of the resident oral microbiome. For example, *Candida* (*C*.) *albicans*, the most common fungal colonizer in the mouth, interacts with oral bacteria. Various oral bacteria, such as *Staphylococcus aureus* and *S. mutans*, stick to *C. albicans* in oral biofilms and can control its pathogenicity. Furthermore, because the presence of *C. albicans* can affect how the bacterial microbiota behaves, these interactions are defined as being multidirectional [16,17,18]. As a result, the sum of all the microorganisms’ interactions may lead to the extreme complexity associated with these multispecies oral biofilms. It has been discovered that *C. albicans* interacts with *Porphyromonas (P.) gingivalis*, a significant etiological factor in chronic periodontitis [16]. This microbial complex influenced human immunity by reducing fibroblast and macrophage responses, whereas cytokine and chemokine significantly decreased compared to pure bacterial infection. The fibroblasts obtained from patients with severe periodontitis were less prone to fungus colonization, showing that the predominating bacterial infection modified the host environment [19]. We draw the conclusion that a milder inflammation at the site of infection may result from the co-infection of *P. gingivalis* with *C. albicans.*

Viruses such as SARS-CoV-2 can directly or indirectly infect mouth tissues of either the mucosa or salivary glands. SARS-CoV-2 invades the host tissues by a helper entry receptor called angiotensin I-Converting Enzyme 2 (ACE2) and transmembrane serine protease 2 (TMPRSS2) [20,21]. Recent studies have documented the expression of *ACE2* found at a high level within several oral epithelial clusters such as salivary glands (SG) ducts, SG serous and SG mucous acini clusters. In addition, *TMPRSS2* was detected to be enriched in the SG epithelia. These results imply that a variety of oral-epithelial-cell subtypes are prone to infection, and they may transfer the coronavirus by their ciliary characteristic [21,22,23,24]. Therefore, since these two receptors are found at a high level in the oral tissue, they may serve as a vehicle for SARS-CoV-2. As a result, TMPRSS2 and ACE2 are traversing and establishing the route of SARS-CoV-2 transmission from the mouth to the respiratory system [21,22,23]. In addition, Huang and his colleagues reported that these two proteins, TMPRSS2 and ACE2, are detected within the lung in COVID-19 patients [24]. Moreover, some studies have used animal models, such as macaque, to study the SARS-CoV-2 infection mechanism [25,26]. They used a combination of intratracheal and intranasal methods to administer a SARS-CoV-2 strain to young and old cynomolgus macaques, resulting in an illness that resembled COVID-19 [25,26]. SARS-CoV-2 RNA was detected in numerous tissues from the tracheobronchial lymph nodes, ileum, and respiratory tract during the autopsies of four macaques. Acute or more severe diffuse alveolar damage (DAD)—which includes alveolar edema; epithelial necrosis; the formation of hyaline membranes; and an accumulation of neutrophils, macrophages, and lymphocytes—was present in two (old) out of four macaques, suggesting that age may play an important role in severe COVID-19 disease [25,26]. It seems that the nasal route is one of the transmission routes of SARS-CoV-2 in macaque. Another study has been carried out on ferrets, and ferrets were susceptible to SARS-CoV-2 infection and might spread the disease through direct or indirect contact (airborne transmission) with other ferrets. The lungs, trachea, gut, and kidney tissues of ferrets all had viral RNA. Importantly, the bronchial epithelium, bronchial lumen, and alveolar wall of infected ferrets showed increased immune infiltration and cellular debris, suggesting that SARS-CoV-2 infection could lead to acute bronchiolitis in ferrets [27]. We suggest that the route of SARS-CoV-2 transmission in macaques and ferrets is similar to the human: starting from the nasal and oral cavity to the respiratory system.

Some cross-sectional studies have suggested that patients with COVID-19 show dysbiosis of the oral microbiome. Scientists noticed a decrease in the diversity of oral microbiomes is associated with the severity of COVID-19. In addition, the oral microbiome is correlated with a decreased level of IgA, which is the first line of defense against bacterial or viral infection, and increases the level of inflammatory cytokines [1,24,28]. Moreover, Bezstarosti et al. found that the severity of COVID-19 is correlated with upregulation of heparin cofactor II [29]. We conclude that COVID-19 may manipulate the immune system of the host since heparin cofactor II plays an important role in immune response. It has been linked to leukocyte-mediated protein breakdown, which causes neutrophils and monocytes to produce cytokines during an inflammatory response [29].

One proteome study has been performed to identify specific interactions between proteins in SARS-CoV-2 and human proteomes. Gordon and his colleagues were able to detect 332 protein interactions between proteins in the SARS-CoV-2 and human proteomes using affinity purification mass spectrometry (AP-MS) [30]. Another proteome study reported that the differential regulation of 27 proteins identified by liquid chromatography with tandem mass spectrometry (LC-MS/MS) was also related to the severity of COVID-19 [31]. In addition, the cross-linking method has been utilized to structurally investigate proteins within SARS-CoV-2, including: Nsp1, Nsp2, and nucleocapsid (N) proteins, to determine full-length atomic models. The full-length Nsp2 could be represented by a single reliable all-atom model according to the study, and its cross-links revealed a complex topology with long-range interactions. The replication–transcription complex has three putative metal-binding sites which are indicative of Nsp2’s function in regulating zinc, according to the model. For the N protein, many intra- and interdomain cross-links were found [32]. Shen et al.’s study used Tandem Mass Tag pro, a chemical tag used for the detection of proteins in samples, combined with (LC-MS/MS), to determine which apolipoproteins in COVID-19-positive blood serum samples are regulated abnormally. Apolipoprotein A1 (APOA-I) and Apolipoprotein M (APOM) are significant in severe COVID-19 cases [33]. Therefore, these proteomic studies could help in the diagnosis and treatment of COVID-19. The cross-linking method offers important stereochemical and electrostatic data on proteins, as well as structural information to emphasize their significance in the cellular context. We conclude that the identification of these case-severity-related proteins/protein-related entities could control the potential severity of COVID-19 cases before the disease develops into a severe case.

One study found an increase in the diversity of oral microbiomes in patients with periodontal diseases and dental caries. On the other hand, there is a decrease in the diversity of oral microbiomes in patients with smoking habits and oral cancer [22,34,35]. Therefore, oral pathogenic bacteria such as the status of periodontal disease and dental caries are affected by the diversity of oral microbiome. In addition, one study was performed in Nigeria on children with HIV. They found that HIV infection and treatment can affect the oral microbiome, which leads to an increase in the state of dental caries in a child [35]. However, the mechanism is not fully understood. In addition, the effect of SARS-CoV-2 on the diversity of the oral microbiome remains to be fully understood. We suggest that SARS-CoV-2 may increase the level of pathogenic bacteria and decrease the level of useful bacteria with poor oral hygiene. As a result, it could cause weakness in the immune system in the body.

The oral cavity acts as a gate and exit for numerous respiratory infections, such as SARS-CoV-2 infection. Usually, SARS-CoV-2 is detected within saliva samples and an abundance of ACE2 receptors in the epithelial cavity [22]. The correlation between pathogenic oral microbiomes and the pathogenesis of respiratory infections has been investigated and several mechanisms have been presented. For example, *Porphyromonas* (*P.*) *gingivalis* is noticed at a high level in patients suffering from periodontitis; this results in lung infections by the aspiration of *P*. *gingivalis* into the lungs. Moreover, periodontal pathology results in modification of the oral mucous membranes made by inflammation [36,37]. In addition, periodontal diseases and chronic systematic diseases such as diabetes and hypertension have been identified as risk factors for severe SARS-CoV-2 infections. There may be a connection between COVID-19 and the complications of periodontitis since the cytokine storm brought on by the COVID-19 infection is very similar to the cytokine imbalance involved in the onset of periodontitis. Chemokines are responsible for the recruitment of inflammatory cells in both COVID-19 and periodontitis [38]. As a result, the pathogenic microbiome could be more subject to adhesion and colonization by pathogenic species of the respiratory tract, followed by their aspiration into the lungs.

Some studies reported that there is a relationship between SARS-CoV-2 infection, oral microbiomes, and oral hygiene. The role of TMPRSS2 is to break the S protein of the SARS-CoV-2 in order for it to fuse with the host cell [36,37]. Besides TMPRSS2 in the oral cavity, the S protein of the SARS-CoV-2 can also be cleaved by pathogenic bacteria found in the oral cavity such as periodontal pathogens able to produce proteases to cleave the S protein of SARS-CoV-2 [2,13,34,35]. Periodontal pathogens are increased by poor oral hygiene, and these pathogens can enhance the expression of ACE2, boost pro-inflammatory cytokines, and degrade the S protein [36]. The penetration and infectivity of SARS-CoV-2 may be increased by the microbial proteases’ destruction of the S protein [34]. COVID-19 can be made worse by neglecting dental hygiene and aspirating periodontal infections [36]. Therefore, pathogenic bacteria in the oral cavity can interfere with TMPRSS2 and ACE to facilitate the entry of SARS-CoV-2 into the host cell. We can conclude that there is a direct relationship between SARS-CoV-2 infection, oral microbiome, and oral hygiene. In addition, maintaining good periodontal health may reduce the host’s vulnerability to COVID-19 and help individuals with the infection avoid COVID-19 aggravation.

Zheng et al. found that in most severe COVID-19 patients, there was an increase in neutrophils and a decrease in lymphocytes, which is crucial in the immune response to viral species. This is an abnormal condition for viral infection. They expected the reason for the increased level of neutrophils to be due to bacterial co-infection and a decreased level of lymphocytes due to a dysfunction of the lymphocytes or the evasiveness of bacterial co-infection [39]. Another study confirmed that 50% of COVID-19 patients, who died, had a secondary bacterial infection [8]. In addition, one study showed the probable effect of oral bacteria in COVID-19 co-infections [40]. A recent metagenomic analysis confirmed a high level of cariogenic and periodontopathogenic bacteria such as *Fusobacterium* in COVID-19 patients [41]. These studies suggested that there is a correlation between the oral microbiome and SARS-CoV-2. It seems that poor oral hygiene may increase the risk factor for COVID-19 complications.

Improving the role of oral health and spreading awareness of oral hygiene could help in avoiding SARS-CoV-2 infections. Since most of the bacteria that cause complications in patients with COVID-19 are present in the mouth, adequate oral hygiene could be the best way to prevent and decrease the risk of acquiring a bacterial superinfection [11]. Oral health is very important, especially for ICU patients. One study found that ICU patients with COVID-19, who do not receive any oral care such as tooth brushing or rinsing of the mouth with water, accumulated hundreds of species of bacteria in their oral cavity. Therefore, they contracted both bacterial and viral infections which may lead to death if are not treated efficiently [1]. Our suggestion here is for each patient in the ICU to be provided with a dental assistant to provide the best oral health care and avoid bacterial infection, especially for COVID-19 patients.

To prevent the spreading of COVID-19 in dental clinics, there are various strategies to follow to overcome the viral infection. Based on the recommendations from WHO and ADA [42], we suggest the following strategies.

## 3. Current Strategies to Control COVID-19 in Dental Clinic

### 3.1. Guides for Dentists


Based on the policy of the Center for Disease Control and Prevention (CDCP), any dentists complaining of flu-like symptoms such as sore throat, fever, headache or cough, should not work [43].Hanging several signs at the entrance door and in the waiting area, with instructions for patients, will be a very effective strategy [43]. Figure 2 shows several instructions to avoid spreading COVID-19.Patients must be informed to come alone to avoid increasing number of people in the waiting area [43].Scheduled appointments should be made at a distance from each other to avoid crowd in the waiting area. The dental-health personnel should form a triage for patients to decide if the appointment is important or can be managed at home [43].Any books, magazines, or toys that could be touched must be removed. An infected area or object can transfer the virus easily to the human body since people touch their faces 10–15 times/h [44].All dentists must be vaccinated against the virus as soon as possible, since they are healthcare workers and at a high risk of exposure to the virus [44].


### 3.2. Patient Screening

During the pandemic, intensive dental care should be undertaken. Initially, telescreening suggested screening for any COVID-19 symptoms, a history of visits to any lockdown countries, or any historical contact with COVID-19 patients. A physician consultation should be made for either active COVID-19 patients or those having recovered from COVID-19. According to the urgency of the case, dentists can decide to perform or delay the dental-treatment procedure [44].

Recent studies have shown that use of pain evaluation questionnaires by dentists is recommended for the evaluation of the degree of pain. Since the evaluation of pain is different from one patient to another, the American Dental Association encourages dentists to use their experience and professionalism to decide if the patients’ complaint needs urgent treatment or not. To find the right decision, telemedicine such as a doctor on the phone could receive a clear diagnosis picture to evaluate the case [45].

Nevertheless, in the case of COVID-19 patients, emergency treatment that is considered life-threatening must be taken with extreme personal protective equipment (PPE). For example, for unsuspected, critical, recovered, or stable cases of COVID-19, urgent treatment can be performed. However, cosmetic cases can be postponed until the patient’s recovery is confirmed [46].

The waiting area should implement the procedure of social distancing. For example, 2 meters of space between two patients. This information should be hanging in the waiting area and patients must be informed at the entrance gate or when a patient reserves an appointment [46,47].

### 3.3. Inside the Dental Clinic

Inside the dental clinic, there are multiple tools such as the patient chair and devices that should be of concern, especially during a pandemic. Recent studies recommended the use of rubber dams, which isolate the operational area from the rest of the oral cavity, in dental treatment. This prevents the generation of saliva. Moreover, it reduces the transmission of diseases such as AIDS, COVID-19, and hepatitis. Very strong suction is needed with the rubber-dam isolation to ensure the reduction in aerosol production. If a rubber dam is not available, a dentist can use old-fashioned procedures such as hand scalers and carisolv [40,48]. Moreover, following a survey of 614 dental practitioners, Paolone et al. reported that the use of preoperatory mouthwashes and rubber dams increased during the pandemic, whereas aerosolization significantly decreased. The need of devices to reduce aerosol was clearly reported [45]. Infection-control materials such as alcohol hand gel, tissues, and no-touch disposal receptacles must be placed in the clinic, waiting area, and at the entrance gate [40,48].

Disposable PPE should be changed from one patient to another, and it must be used once. There is a specific procedure for dentists to wear and remove PPE which is described by the Centers for Disease Control and Prevention (CDCP) [49]. When wearing PPE, a gown should be followed by a mask, then goggles or a face shield, then gloves. In the case of taking off PPE, gloves should be removed while removing the gown, then goggles or the face shield, then the mask, followed by washing hands directly [49].

Some dental clinics recommend single-use tools (disposable) such as diagnostic kits, syringes, and burs since during the COVID-19 pandemic, the adequate sterilization of dental reusable tools is not accurate. In addition, the long process of sterilization and disinfection may lead dentists and dental staff to cross-contaminate instruments [40].

When patients arrive at the dental clinic, the dentist must ask the patient about any symptoms related to COVID-19 or exposure to anybody with COVID-19. Moreover, checking of the body temperature using a non-contact thermometer should be performed at the entrance of each dental clinic [45]. If a patient has a body temperature above 37 °C, the dental treatment must be postponed, and the patient must be sent for a COVID-19 test and informed to self-quarantine if positive. In addition, the health community authorities should be informed. However, if the case is an emergency, the dental treatment must be conducted in a negative pressure room to prevent the spreading of virus and the dentist must wear personal protective equipment (PPE) such as a protective gown, face mask (N95), gloves and goggles or face shield. If the body temperature is below 37 °C and the patient is asymptomatic, dental treatment could be performed with preventative methods to avoid spreading the virus [45,47,50].

Radiographic imaging is another factor that should be considered by dentists during the pandemic. Recent studies have shown that dentists and radiology technologists were infected with COVID-19 while examining COVID-19 patient. Therefore, it is important to establish guidelines that reduce the risk rate of transmission. Scientists recommend using extraoral imaging techniques such as panoramic, cephalometric, and cone beam computed tomography (CBCT) projections instead of intraoral imaging to reduce the overproduction of saliva, sneezing, or coughing. In addition, these cases could occur in intraoral imaging. Therefore, in some cases, if there is a need for intraoral imaging, it should be followed by an appropriate covering and wrapping of sensors, to avoid cross-contamination [51].

Dentists recommend using anti-retraction handpieces rather than normal handpieces for extra prevention during the COVID-19 pandemic. Anti-retraction handpieces have an anti-retraction valve in the body of the handpiece, which can be autoclaved, considerably decreasing the backflow of oral bacteria and viruses into the non-sterilizable dental unit hoses, which could cause cross-contamination [47,49]. In addition, it is recommended to use resorbable sutures, which are stitches made from biological or artificial materials that the body can naturally absorb over time, thus eliminating the requirement for another appointment and also to decrease contact with patients [49]. Some scientists reported that antimicrobial mouthwash is not effective to eliminate SARS-CoV-2. It is effective to decrease the number of oral microbes, as indicated in the Diagnosis and Treatment of Novel Coronavirus Pneumonia Guidelines (the 5th edition) declared by The National Health Commission of the People’s Republic of China, [49,50]. However, Peng and colleagues found that using mouthwashes, which contained 0.2% povidone and 1.5% hydrogen peroxide, seems to be efficient at reducing the number of microbes as well as SARS-CoV-2 elimination [46].

At the end of the day, the last dental appointment must be booked by the Aerosol Generating Dental Operations (AGDP) to avoid cross-contamination. If, by accident, a dentist has contact with a COVID-19 patient without an N95 mask, the dentist is at a risk based on the information from the CDC. It is recommended to perform the COVID-19 test immediately. If the test is positive, the dentist must quarantine for 14 days. In addition, all treated patients or colleagues who had contact with the infected dentist must be informed [49,51].

### 3.4. Waste Disposal Management

All disposable PPE should be discarded properly and cleaned with water and soap. The manufacturer’s instructions for disinfecting must followed for all disposable dental equipment. For example, for handpieces, cleaning and heat sterilization are the recommended methods by the CDC. However, handpieces that cannot be sterilized by any of the recommended procedures should not be used [52,53]. Ultrasonic scalers and air-water syringes must be sterilized by autoclave after using for each patient, to ensure that all microbes are killed [54].

One study has summarized the steps to deal with biomedical or dental waste as the following: first, the collection of biomedical waste should happen three times a day [54], i.e., in the morning, in the afternoon, and in the evening, to decrease the level of infection because biomedical waste remains in the clinic for less than 3 hours. Second, biomedical waste should be transported to the central storage facility by an appropriate vehicle to burn it and then dispose it in a licensed landfill. Third, the container with infectious waste should be marked with the international biohazard symbols. Common waste such as Mercury Amalgam must be collected, transported, and discarded separately. Sharp objects should be separated inside tamper-proof and puncture-proof containers made of dense cardboard supplied with a plastic lining [4,53]. These steps are very easy to follow to reduce the level of infection, since there are several diseases (AIDS, Hepatitis, and COVID-19) that are transported to humans in the absence of awareness of biomedical-waste handling.

Some studies recommended using environmentally friendly materials to reduce medical-waste-related environmental health risks, since dental waste will negatively impact the environment if every dentist does not handle it correctly [52]. Sabbahi and Goriuc’s study recommends that in order to maintain protection for the people and environment, biomedical waste should be separated and eliminated in a safe manner [53].

In recent years, there has been a considerable decrease in awareness of biomedical-waste management worldwide. Medical waste is a major concern, especially in dental clinics, because of its high risk of cross-contamination. According to WHO, biomedical waste such as human tissue, blood, and saliva should not be stored in the clinic for more than 3 hrs. The WHO released well-organized guidelines regarding the handling and management of waste products in dental clinics or hospitals (Figure 3). It has classified biomedical waste into seven types and each one has a specific bag color: general waste, pathological waste, sharps, infectious waste, chemical waste, radioactive waste, and pharmaceutical waste [42,49].

## 4. Conclusions and Future Directions

In conclusion, there are two main transmission routes of COVID-19: oral and nasal. The oral cavity has a mixture of several species of the oral microbiome, which is a suitable environment to grow bacteria and viruses. Most studies prove that awareness of oral hygiene is key to preventing oral diseases and may lower the transmission rate of COVID-19. In addition, infection control at the dental clinic is mandatory because everbody at the clinic needs infection control. For example, patients should be educated about infection control, and how they can play their part in preventing the spread of infection from one patient to another. Dentists are responsible for educating and training their team members regarding the importance of cross-infection control. They must also ensure the professional development of the newcomers in their practice. In addition, dental-healthcare professionals should keep their knowledge up to date to educate patients and staff. In addition, dental-health personnel should follow the instructions from WHO.

Future studies should investigate whether there are any other transmission routes of COVID-19. In addition, is there an emergency action plan for PPE shortages? Is it logical to rely on a degree of pain using a doctor on phone as the criterion to classify dental emergencies? The mechanism of how oral microbiomes influence COVID-19 should be under investigation. Therefore, it is necessary to better understand the impact of COVID-19 on the oral microbiome. It is very important to have answers for the previous questions in order to obtain the best view of the pandemic.

## Figures and Tables

**Figure 1 microorganisms-11-00146-f001:**
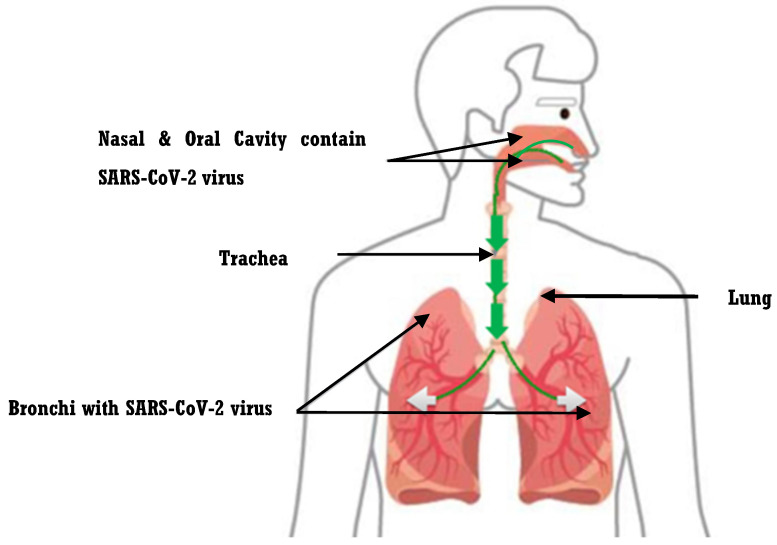
**Diagram showing the route of SARS-CoV-2 transmission.** Based on the information of the previous study performed on Syrian hamster model, which reported that the pathogenic of SARS-CoV-2 within the saliva or nasal cavity can be transferred into the lower respiratory tract and cause severe infection [11,12]. Here, we assume the same may happen in the case of COVID-19.

**Figure 2 microorganisms-11-00146-f002:**
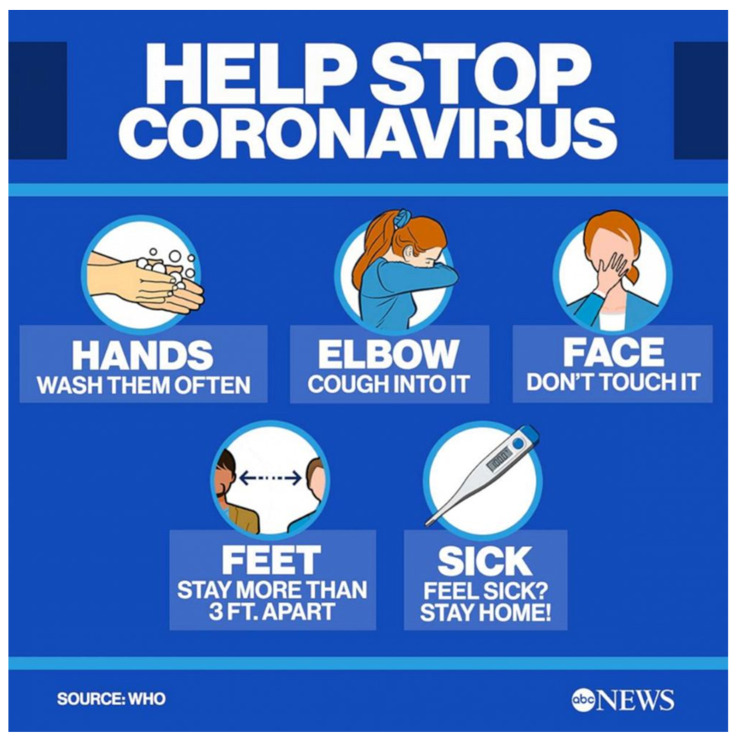
How to stop coronavirus (WHO).

**Figure 3 microorganisms-11-00146-f003:**
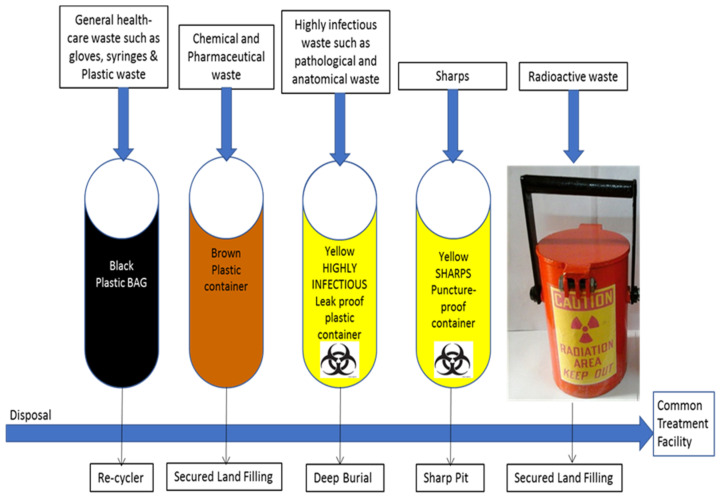
The recommended coding color for biomedical waste and the disposal procedure for each waste type [42].

## Data Availability

Not applicable.

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
