# Peer review of "An Outlook on Dental Practices to Avoid the Oral Transmission of COVID-19"

_microorganisms, 2023, doi:10.3390/microorganisms11010146_

Round 1

Reviewer 1 Report

Dear authors,

thanks for submitting to “Microorganism” the paper entitled:

An Outlook on Dental Practices to Avoid Oral Transmission of 2 

COVID-19 Infection”

the topic is original and relevant in the field. It does address a specific gap in the field especially because Covid-19 pandemic is not over.

Nevertheless, here are some suggestions to improve your manuscript.

Lines 129-130:

Please add a reference to support the sentence:

“That’s why people, who have a high level of saliva production, seem to have healthy teeth and low dental caries. ”

Lines 166-167:

The authors wrote:

“Many cross-sectional studies have suggested that patients with COVID-19 show dysbiosis of the oral microbiome ”

Please add the reference of these many cross-sectional studies.

If the authors are referring to references: 1,23,24 the authors should define them “Some cross-sectional studies”

Line 198:

Please do not refer to “Scientists”. Use the name of the researchers. In this case: Zheng M. et al.

Lines 232-233: The authors could add a link to allow the readers to download a high-resolution pdf image to use in their clinics.

Lines 255: please change “is” to “has”

Lines 271-7

The authors could emphasize even more the risks of aerosol generation during the Pandemic outbreak.

A sentence that could be added could be the following:

“Paolone et al., in a survey of 614 dental professionals, reported that increased use of preoperatory mouthwashes and rubber dam was referred during the pandemic, while aerosolization (i.e. ultrasound) was drastically reduced. The need of devices to reduce aerosol was clearly reported.”

Paolone G, Mazzitelli C, Formiga S, Kaitsas F, Breschi L, Mazzoni A, Tete G, Polizzi E, Gherlone E, Cantatore G. 1 year impact of COVID-19 pandemic on Italian dental professionals: a cross-sectional survey. Minerva Dent Oral Sci. 2021 Dec 1. https://doi.org/10.23736/S2724-6329.21.04632-5 . PMID: 34851068.

Line 338:

Please add a space inthe middle of: proceduresshould 

Author Response

Manuscript ID: microorganisms-2118807

Title: An Outlook on Dental Practices to Avoid Oral Transmission of COVID-19

Infection

Authors: Waad Kattan, Lama Alsamadani, Ghadah Alahmari, Wasan Aljuhani, Maha

Almabadi, and Manal Alsulami *

Dear Microorganisms (MDPI) peer review Team,

Enclosed please find a resubmission of the above manuscript. We are pleased to inform you that we have addressed all the points raised by the reviewers. We would like to thank the reviewers for their helpful comments and hope that you will consider the modified manuscript for publication in MDPI in Microbiology of Oral Diseases. Please do not hesitate to contact me by email Manno552010@yahoo.com or malsulami@vision.edu.sa if you have any queries with regards to the manuscript.

Yours sincerely,

Dr. Manal Alsulami

Reviewer #1 Comments:

General Comment: the topic is original and relevant in the field. It does address a specific gap in the field especially because Covid-19 pandemic is not over. Nevertheless, here are some suggestions to improve your manuscript.

Response: We really appreciate your support of the purpose of this manuscript. We carefully read all of your comments and have done our best to respond to each one individually. After this revision, we hope that the manuscript has been raised to MDPI in Microbiology of oral diseases standards.

  • Lines 129-130: Please add a reference to support the sentence:

“That’s why people, who have a high level of saliva production, seem to have healthy teeth and low dental caries ”

Response: Thank you for pointing this out. We made the required changes as recommended. Please find it in line 116-117.

  • Lines 166-167: The authors wrote: “Many cross-sectional studies have suggested that patients with COVID-19 show dysbiosis of the oral microbiome ”

Please add the reference of these many cross-sectional studies.

If the authors are referring to references: 1,23,24 the authors should define them “Some cross-sectional studies”

Response: Thank you for pointing this out. We made the required changes as recommended. Please find it in line 171.

  • Line 198: Please do not refer to “Scientists”. Use the name of the researchers. In this case: Zheng M. et al.

Response: Thank you for pointing this out. We made the required changes as recommended. Please find it in line 247.

  • Lines 232-233: The authors could add a link to allow the readers to download a high-resolution pdf image to use in their clinics.

Response: Thank you for pointing this out. We made the required changes as recommended. Please find it in line 279.

  • Lines 255: please change “is” to “has”

 Response: Thank you for pointing this out. We made the required changes as recommended. Please find it in line 297.

  • Lines 271-7: The authors could emphasize even more the risks of aerosol generation during the Pandemic outbreak. A sentence that could be added could be the following:

“Paolone et al., in a survey of 614 dental professionals, reported that increased use of preoperatory mouthwashes and rubber dam was referred during the pandemic, while aerosolization (i.e. ultrasound) was drastically reduced. The need of devices to reduce aerosol was clearly reported.” Paolone G, Mazzitelli C, Formiga S, Kaitsas F, Breschi L, Mazzoni A, Tete G, Polizzi E, Gherlone E, Cantatore G. 1 year impact of COVID-19 pandemic on Italian dental professionals: a cross-sectional survey. Minerva Dent Oral Sci. 2021 Dec 1. https://doi.org/10.23736/S2724-6329.21.04632-5 . PMID: 34851068.

Response: Thank you for pointing this out. We made the required changes as recommended. Please find it in line 325-328.

  • Line 338: Please add a space in the middle of: proceduresshould

Response: Thank you for pointing this out. We made the required changes as recommended. Please find it in line 390.

Reviewer 2 Report

The review aimed to describe the interplay between SARS-CoV-2 infection, oral microbiome, and oral hygiene, as well as the practice to control COVID-19 in dental clinic. However, this review is still very preliminary and has a number of issues that need to be addressed.

Major issues:

1.       The authors should discuss more about the challenge and gap in the relationship between SARS-CoV-2 infection, oral microbiome, and oral hygiene.

2.       The authors should consider comparing the policy and practice to control COVID-19 in dental clinical in different countries and descript their input on those policy and practice.

3.       The authors should cite references for each statement properly, except those proposed in this article. Some of the statements which need references are listed here: Lines 53, 64, 67, 72, 76, 84, 93, 96, 115, 132 and so on.

4.       The authors should use the term of SARS-CoV-2 and COVID-19 properly. For example, in Line 34, SARS-CoV-2 should be a pathogen instead of a respiratory syndrome. In Lines 90 and 101, it should be SARS-CoV-2 instead of COVID-19. The authors should check the terms in the whole article.

5.       The author should make their statements more specific. For example: In Lines 52-53, the authors should give examples of those diseases. In Line 73, the author should describe what the “rest of the lower respiratory tract” indicated. Line 71, what did “most studies” indicate?

6.       The figure legend of Figure 1 is not accurate. First, the figure is not showing the “COVID-19 transmission” but SARS-CoV-2 viruses aspiration form oral cavity to lung. Second, The assumption has already been demonstrated (at least in partial) on an animal model conducted by a HKU team (https://www.sciencedirect.com/science/article/pii/S2666379120301634).

7.       Line 150, there have already been a lot of animal models and organoid studies which tried to address this question. The authors should summary them.

8.       Rewrite L152-164. This paragraph is confusing.

9.       Line 170-171, the statement has already been proposed by a lot of literature. The author should summary those literature.

Minor issues:

1.       The number of COVID-10 related death needs to be updated to a more recent data.

2.       Line 80, the “with” should be within.

3.       All the terms should be referred to by full name when they appear the first time in this article and by abbreviation afterwards. For example: “S. mutans” in Line 113, “PPE” in Line 260.

4.       A comma needs to be added after “As a result” in Line 120.

5.       Line 148, it should be angiotensin.

6.       It should be SARS-CoV-2 instead of “SARSCOV-2” in Lines 157 and 161.

7.       The grammar and spelling need to be double checked for the entire article.

Author Response

Manuscript ID: microorganisms-2118807

Title: An Outlook on Dental Practices to Avoid Oral Transmission of COVID-19

Infection

Authors: Waad Kattan, Lama Alsamadani, Ghadah Alahmari, Wasan Aljuhani, Maha

Almabadi, and Manal Alsulami *

Dear Microorganisms (MDPI) peer review Team,

Enclosed please find a resubmission of the above manuscript. We are pleased to inform you that we have addressed all the points raised by the reviewers. We would like to thank the reviewers for their helpful comments and hope that you will consider the modified manuscript for publication in MDPI in Microbiology of Oral Diseases. Please do not hesitate to contact me by email Manno552010@yahoo.com or malsulami@vision.edu.sa if you have any queries with regards to the manuscript.

Yours sincerely,

Dr. Manal Alsulami

Reviewer #2 Comments:

General Comment: The review aimed to describe the interplay between SARS-CoV-2 infection, oral microbiome, and oral hygiene, as well as the practice to control COVID-19 in dental clinic. However, this review is still very preliminary and has a number of issues that need to be addressed.

Response: We really appreciate your support of the purpose of this manuscript. We carefully read all of your comments and have done our best to respond to each one individually. After this revision, we hope that the manuscript has been raised to MDPI in Microbiology of oral diseases standards.

Major issues:

  1. The authors should discuss more about the challenge and gap in the relationship between SARS-CoV-2 infection, oral microbiome, and oral hygiene.

Response: Thank you for pointing this out. We made the required changes as recommended. Please find it in line 232-246.

  1. The authors should consider comparing the policy and practice to control COVID-19 in dental clinical in different countries and descript their input on those policy and practice.

Response: Thank you for the suggestion. Here in our review paper, the goal is not to compare between different methods in different countries, we are just present the policies and procedures that have been recommended by WHO, Centres for disease control and prevention and American dental association.

  1. The authors should cite references for each statement properly, except those proposed in this article. Some of the statements which need references are listed here: Lines 53, 64, 67, 72, 76, 84, 93, 96, 115, 132 and so on.

Response: Thank you for pointing this out. We made the required changes as recommended. However, sometimes several sentences are having the same references, so instead of repeating the same references after each sentence, we put the citation at the last sentence that belongs to the same references. Therefore, all the sentences before the last one will have the same citation.

  1. The authors should use the term of SARS-CoV-2 and COVID-19 properly. For example, in Line 34, SARS-CoV-2 should be a pathogen instead of a respiratory syndrome. In Lines 90 and 101, it should be SARS-CoV-2 instead of COVID-19. The authors should check the terms in the whole article.

Response: Thank you for pointing this out. We made the required changes as recommended.

  1. The author should make their statements more specific. For example: In Lines 52-53, the authors should give examples of those diseases. In Line 73, the author should describe what the “rest of the lower respiratory tract” indicated. Line 71, what did “most studies” indicate?

Response: Thank you for pointing this out. We made the required changes as recommended. Please find it in lines 52-53, line 59, and lines 57-58.

  1. The figure legend of Figure 1 is not accurate. First, the figure is not showing the “COVID-19 transmission” but SARS-CoV-2 viruses aspiration form oral cavity to lung. Second, The assumption has already been demonstrated (at least in partial) on an animal model conducted by a HKU team (https://www.sciencedirect.com/science/article/pii/S2666379120301634).

         Response: Thank you for pointing this out. We made the required changes as recommended.  Please see Figure 1.

  1. Line 150, there have already been a lot of animal models and organoid studies which tried to address this question. The authors should summary them.

Response: Thank you for pointing this out. We made the required changes as recommended. Please find it in lines 154-170.

  1. Rewrite L152-164. This paragraph is confusing.

Response: Thank you for pointing this out. We made the required changes as recommended. Please find it in lines 141-153.

  1. Line 170-171, the statement has already been proposed by a lot of literature. The author should summary those literature.

Response: Thank you for pointing this out. We made the required changes as recommended. Please find it in lines 181-203.

Minor issues:

  • The number of COVID-10 related death needs to be updated to a more recent data.

Response: Thank you for pointing this out. We made the required changes as recommended. Please find it in line 28.

  • Line 80, the “with” should be within.

Response: Thank you for pointing this out. We made the required changes as recommended. Please find it in line 66.

  • All the terms should be referred to by full name when they appear the first time in this article and by abbreviation afterwards. For example: “S. mutans” in Line 113, “PPE” in Line 260.

Response: Thank you for pointing this out. We made the required changes as recommended. Please find it in lines 128 & 309.

  • A comma needs to be added after “As a result” in Line 120.

Response: Thank you for pointing this out. We made the required changes as recommended. Please find it in line 131.

  • Line 148, it should be angiotensin.

Response: Thank you for pointing this out. We made the required changes as recommended. Please find it in line 143.

  • It should be SARS-CoV-2 instead of “SARSCOV-2” in Lines 157 and 161.

Response: Thank you for pointing this out. We made the required changes as recommended. Please find it in lines 150 & 151.

  • The grammar and spelling need to be double checked for the entire article.

Response: Thank you for pointing this out. We made the required changes as recommended. Also, we sent the whole article to the English department at Vision Medical College for grammar checking.

Reviewer 3 Report

The originality of the subject is lacking. The use of English is poor, a specialist should be consulted. Please see the enclosed pdf for a point-by-point analysis.

Author Response

Manuscript ID: microorganisms-2118807

Title: An Outlook on Dental Practices to Avoid Oral Transmission of COVID-19

Infection

Authors: Waad Kattan, Lama Alsamadani, Ghadah Alahmari, Wasan Aljuhani, Maha

Almabadi, and Manal Alsulami *

Dear Microorganisms (MDPI) peer review Team,

Enclosed please find a resubmission of the above manuscript. We are pleased to inform you that we have addressed all the points raised by the reviewers. We would like to thank the reviewers for their helpful comments and hope that you will consider the modified manuscript for publication in MDPI in Microbiology of Oral Diseases. Please do not hesitate to contact me by email Manno552010@yahoo.com or malsulami@vision.edu.sa if you have any queries with regards to the manuscript.

Yours sincerely,

Dr. Manal Alsulami

Reviewer #3 Comments:

General Comment: The originality of the subject is lacking. The use of English is poor, a specialist should be consulted. Please see the enclosed pdf for a point-by-point analysis.

Response: We really appreciate your support of the purpose of this manuscript. We carefully read all of your comments and have done our best to respond to each one individually. After this revision, we hope that the manuscript has been raised to MDPI in Microbiology of oral diseases standards.

  • The introduction is too long and repetitive.

Response: Thank you for pointing this out. We made the required changes as recommended. We delete one paragraph from lines 57-70.

  • Reformulate not clear lines 144-145.

Response: Thank you for pointing this out. We made the required changes as recommended. Please find it in lines 131-140.

  • The authors should discuss the interaction between periodontitis and COVID-19.

Response: Thank you for pointing this out. We made the required changes as recommended. Please find it in lines 224-229.

  • Correction: dentists is complaining of.

Response: Thank you for pointing this out. We made the required changes as recommended. Please find it in line 276.

  • Correction: Signs

Response: Thank you for pointing this out. We made the required changes as recommended. Please find it in line 278.

  • What is PPE?

Response: Thank you for pointing this out. We made the required changes as recommended. Please find it in line 309.

  • Critical or non-critical?

Response: Thank you for pointing this out. We mean critical because we are giving an example of emergency treatment. Therefore, critical case of COVID-19 needs an emergency treatment. Please find it in line 310.

Round 2

Reviewer 2 Report

The manuscript has been improved a lot after revision, but there are still some issues:

1) L34, pathogenic virus instead of pathogen virus;

2) Figure 1 still need improvement, e.g., the dots on the top right need to be removed.

3) The use of full name and abbreviation of the bacteria is still not consistent in the manuscript, e.g., L185, Streptococcus (S.) mutans instead of Streptococcus (S) mutans; L225, abbreviation should be used for Streptococcus mutans and so on. 

4) L249, TMPRSS2 is not a receptor.

5) L250-266: The authors did not make a clear point. It seems the two animal studies do not suppot their statement.

6) L259, how to define "extremely" susceptible?

7) L463, the authors should double check if the ACE2 can break S protein?

8) L466, the author should consider describing details how bacteria directly cleavage the S protein

Author Response

Manuscript ID: microorganisms-2118807

Title: An Outlook on Dental Practices to Avoid Oral Transmission of COVID-19

Infection

Authors: Manal Alsulami*, Waad Kattan, Lama Alsamadani, Ghadah Alahmari, Wasan Aljuhani, Maha

Almabadi

Dear Microorganisms (MDPI) peer review Team,

Enclosed please find a resubmission of the above manuscript. We are pleased to inform you that we have addressed all the points raised by the reviewers. We would like to thank the reviewers for their helpful comments and hope that you will consider the modified manuscript for publication in MDPI in Microbiology of Oral Diseases. Please do not hesitate to contact me by email Manno552010@yahoo.com or malsulami@vision.edu.sa if you have any queries with regards to the manuscript.

Yours sincerely,

Dr. Manal Alsulami

Reviewer #1 Comments:

General Comment: The manuscript has been improved a lot after revision, but there are still some issues:

Response: We really appreciate your support of the purpose of this manuscript. We carefully read all of your comments and have done our best to respond to each one individually. After this revision, we hope that the manuscript has been raised to MDPI in Microbiology of oral diseases standards.

  • L34, pathogenic virus instead of pathogen virus

Response: Thank you for pointing this out. We made the required changes as recommended. Please find it in line 35.

  • Figure 1 still need improvement, e.g., the dots on the top right need to be removed. Response: Thank you for pointing this out. We made the required changes as recommended. Please find it in Fig. 1.
  • The use of full name and abbreviation of the bacteria is still not consistent in the manuscript, e.g., L185, Streptococcus (S.) mutans instead of Streptococcus (S) mutans; L225, abbreviation should be used for Streptococcus mutans and so on.

Response: Thank you for pointing this out. We made the required changes as recommended. Please find it in lines 119, 164, & 264.

  • L249, TMPRSS2 is not a receptor.

 Response: Thank you for pointing this out. We made the required changes as recommended. Please find it in lines 166-168, and 189.

  • L250-266: The authors did not make a clear point. It seems the two animal studies do not support their statement.

Response: Thank you for pointing this out. We made the required changes as recommended. Please find it in line 198-203.

  • L259, how to define "extremely" susceptible?

Response: Thank you for pointing this out. We made the required changes as recommended. Please find it in line 201. The meaning of extremely in this sentence was as very.

  • L463, the authors should double check if the ACE2 can break S protein?

Response: Thank you for pointing this out. We made the required changes as recommended. Please find it in lines 277-278.

  • L466, the author should consider describing details how bacteria directly cleavage the S protein?

Response: Thank you for pointing this out. We made the required changes as recommended. Please find it in lines 279-281.

Reviewer 3 Report

The manuscript has been improved

Author Response

Manuscript ID: microorganisms-2118807

Title: An Outlook on Dental Practices to Avoid Oral Transmission of COVID-19

Infection

Authors: Manal Alsulami*, Waad Kattan, Lama Alsamadani, Ghadah Alahmari, Wasan Aljuhani, Maha

Almabadi

Dear Microorganisms (MDPI) peer review Team,

Enclosed please find a resubmission of the above manuscript. We are pleased to inform you that we have addressed all the points raised by the reviewers. We would like to thank the reviewers for their helpful comments and hope that you will consider the modified manuscript for publication in MDPI in Microbiology of Oral Diseases. Please do not hesitate to contact me by email Manno552010@yahoo.com or malsulami@vision.edu.sa if you have any queries with regards to the manuscript.

Yours sincerely,

Dr. Manal Alsulami

Comments and Suggestions for Authors:

The manuscript has been improved.

Response: We really appreciate your support of the purpose of this manuscript. We hope that the manuscript has been raised to MDPI in Microbiology of oral diseases standards.